# Spatiotemporal Niche Separation among Passeriformes in the Halla Mountain Wetland of Jeju, Republic of Korea: Insights from Camera Trap Data

**DOI:** 10.3390/ani14050724

**Published:** 2024-02-26

**Authors:** Young-Hun Jeong, Sung-Hwan Choi, Maniram Banjade, Seon-Deok Jin, Seon-Mi Park, Binod Kunwar, Hong-Shik Oh

**Affiliations:** 1Interdisciplinary Graduate Program in Advanced Convergence Technology and Science, Jeju National University, Jeju-si 63243, Jeju-do, Republic of Korea; sung_1220@naver.com (S.-H.C.); kunwarbinod22@gmail.com (B.K.); 2National Institute of Ecology, 1210 Geumgang-ro, Maseru-myeon, Seocheon-gun 33657, Chungcheongnam-do, Republic of Korea; mani88zoo@gmail.com (M.B.); withbirds@nie.re.kr (S.-D.J.); 3Research Institute for Basic Sciences, Jeju National University, Jeju-si 63243, Jeju-do, Republic of Korea; 4Faculty of Science Education, Jeju National University, Jeju-si 63243, Jeju-do, Republic of Korea

**Keywords:** behavior ecology, competition, community structures, camera trap, habitat structure

## Abstract

**Simple Summary:**

Our comprehensive study using camera traps in the Halla Mountain Wetland offers significant insights into the temporal activities and habitat preferences of Passerine birds. Key findings include distinct morning activities in species such as *Cyanoptila cyanomelana* and *Horornis canturians* and afternoon peaks in others such as *Muscicapa dauurica*. We observed a preference for wetlands and an underutilization of grasslands, indicating complex species–habitat interactions. Additionally, our study highlights the impact of population density on habitat usage and reveals substantial variations in seasonal activity among different species, demonstrating their adaptive strategies. This research contributes to a deeper understanding of the ecological dynamics and niche partitioning within these bird communities. However, the fixed field of view and intermittent recording capability of camera traps can miss certain activities, and external factors such as weather and human presence can influence the observed behavior. While providing valuable insights, this study underscores the need to integrate camera trap data with other observational methods for a more comprehensive understanding of avian behavior.

**Abstract:**

This study analyzed 5322 camera trap photographs from Halla Mountain Wetland, documenting 1427 independent bird sightings of 26 families and 49 species of Passeriformes. Key observations include morning activities in *Cyanoptila cyanomelana* and *Horornis canturians* and afternoon activity in *Muscicapa dauurica* and *Phoenicurus auroreus*. Wetlands were significantly preferred (P_i = 0.398) despite their smaller area, contrasting with underutilized grasslands (P_i = 0.181). Seasonal activity variations were notable, with overlap coefficients ranging from 0.08 to 0.81 across species, indicating diverse strategies in resource utilization and thermoregulation. Population density was found to be a critical factor in habitat usage, with high-density species showing more consistent activity patterns. The study’s results demonstrate the ecological adaptability of Passeriformes in the Halla Mountain Wetland while highlighting the limitations of camera trapping methods. These limitations include their fixed field of view and intermittent recording capability, which may not fully capture the spectrum of complex avian behaviors. This research underlines the need for future studies integrating various methodologies, such as direct observation and acoustic monitoring, to gain a more comprehensive understanding of avian ecology.

## 1. Introduction

Wetlands serve as vital bird habitats, providing essential resources such as nesting sites, food, and shelter [1,2]. These habitats harbor a rich diversity of bird species and contribute to the overall ecological balance of the region [3]. Wetlands are particularly influential in shaping the daily and seasonal activity patterns of avian species. For instance, the availability of nesting sites and food resources in wetlands can directly affect how birds allocate their time to various activities throughout the day and year. Additionally, the structured environment within wetlands not only influences various ecological interactions, such as competition and predation, but also ecological factors such as resource availability, which influences the behavior and survival of these birds [4,5]. Therefore, understanding the intricate relationship between wetland ecosystems and the behavioral ecology of birds is crucial to comprehending the complexity of their interaction patterns and developing effective conservation strategies [6,7].

Competitive interactions among bird species are crucial in determining the structure and function of avian communities [8]. These interactions affect resource use, habitat selection, and, ultimately, the survival and reproduction of individual species [9]. Interspecies competition can lead to niche partitioning, where species adapt their behaviors and activity times to reduce competition and coexist within the same habitat [10,11]. Therefore, examining competitive interactions in the context of circadian rhythms can help us better understand the complex relationships among bird species and their environment [12].

In dynamic ecosystems, the concepts of niche separation and overlap are pivotal in understanding avian diversity and behavior. Niche separation refers to the process where bird species evolve distinct behaviors, feeding times, or habitat preferences, reducing direct resource competition. This separation allows for diverse species to coexist in the same ecological space [13]. Conversely, niche overlap occurs when species share similar ecological roles or resources, leading to increased competition. This overlap can drive evolutionary changes as species adapt to minimize competition or face possible exclusion [14,15]. 

Camera trapping has long been an effective tool for determining species distribution, estimating population density, and facilitating various research dimensions, such as studying animals’ circadian rhythms and constructing biodiversity databases [16]. While camera trapping techniques historically focused on the spatial elements of species ecology and population dispersion, recent shifts in research focus have occurred due to the capacity of timestamped photographs to reveal temporal changes in species’ behavioral patterns [17]. A key area of research in camera trapping and temporal data is the observation of wildlife behavioral patterns. To ensure the precision of results derived from these methods, it becomes essential to leverage a diverse dataset that includes time stamps that correspond to the data [18,19,20,21,22].

We hypothesized that diverse ecological interactions, which may include niche separation and habitat characteristics, would shape bird activity patterns. Furthermore, we anticipated that habitat characteristics and environmental factors would also mediate the relationship between bird activity patterns.

Our primary objectives were the following: (1) Identify the bird species in the wetland. (2) Determine their activity patterns in relation to their circadian rhythms. (3) Explore species interactions shaping these activity patterns. 

## 2. Materials and Methods

### 2.1. Study Site

Hallasan National Park is an exceptional and ecologically significant area on Jeju Island, Republic of Korea. It has received several designations, including UNESCO Biosphere Reserve, World Natural Heritage site, and Global Geopark, highlighting its importance for conservation and biodiversity [23]. The study was conducted in a high-altitude wetland situated at an elevation of 950 m within Hallasan National Park (33°22′13.97″ N, 126°26′54.76″ E). This wetland is ephemeral in nature, remaining dry and filling up with water only during rainfall periods. It was selected for investigation due to its underexplored status and potential to offer insights into bird species interactions and their adaptation to dynamic environmental conditions. The study site is home to more than 70 bird species [24], including the vulnerable *Pitta nympha* and near threatened *Terpsiphone atrocaudata*. The vegetation surrounding the wetland is predominantly composed of broad-leaved forests, such as *Carpinus laxiflora*, *Styrax japonicus*, and *Quercus serrata*, with mixed forests that include coniferous species (Figure 1). Throughout the study period, the highest recorded temperature was 28.6 °C, the lowest was −20.2 °C, and the average precipitation was 563.02 mm [25]. The study site has experienced minimal anthropogenic disturbance, ensuring that bird species interactions can be studied in a relatively pristine environment.

### 2.2. Filed Sampling 

A total of 24 camera traps (ROBOT D30, Bushwhacker Shenzhen, China) were deployed throughout the study site from March 2018 to March 2023. The camera traps operated daily from 5 AM to 7 PM, covering a significant portion of the day and night cycle. Out of the 24 cameras, 5 were installed in wetlands and grasslands, 6 in shrub areas, and 8 in tree areas. The cameras were mounted on supports over 2 m high, except in wetland and grassland areas, where they were placed at heights less than 2 m to better observe the individuals. For tree habitats, the camera supports were installed at heights exceeding 4 m, ensuring an expansive view of the area. To maintain effective coverage and reduce overlap, each camera was placed at least 100 m apart. This setup facilitated the comprehensive monitoring of bird species interactions and activity patterns across different habitats [26]. The cameras captured three consecutive photographs and 10-second videos whenever motion was detected. It was assumed that each camera had a maximum coverage distance of 5 m. This distance is considered the radius of the area that each camera can effectively monitor. Given the circular nature of the camera’s field of view, the area covered by each camera was modeled as a circle. The area of a circle is calculated using the formula A = πr², where ‘A’ represents the area and ‘r’ is the radius of the circle. In this case, the radius ‘r’ is 5 m. To calculate the total coverage area for each ecological zone, the area covered by a single camera was multiplied by the number of cameras in that zone. The calculations were as follows. For wetlands and grasslands, with 5 cameras: Total Area = 5 × π × (5 m)²; for shrublands, with 6 cameras: Total Area = 6 × π × (5 m)²; and for trees, with 8 cameras: Total Area = 8 × π × (5 m)². Passeriformes were analyzed by categorizing them into nesting and foraging guilds based on their nesting and feeding locations, respectively. The nesting guilds were divided into groups that breed in the canopy (C), ground-shrub vegetation (GS), and secondary cavity nesters (S) that utilize existing cavities in trees or other structures [27]. Furthermore, the foraging guilds were classified according to the main foraging locations of forest birds, such as foliage searchers (FS) that explore the canopy for food, ground-shrub foragers (GF) that search for food in the ground and shrub vegetation, and aerial insect pursuers (AI) that catch insects in the air [24,28,29]. It is important to note that the nesting and foraging guilds are general descriptions applicable only to the habits of the species observed in the study area (Appendix A).

### 2.3. Activity Pattern 

The cameras were programmed to follow Korean Standard Time. Consecutive captures of the same species within a one-hour interval were considered redundant and excluded from the analysis. To examine bird activity patterns in relation to circadian rhythms, we categorized observation times into three distinct time zones: morning (5:00–10:00 h), noon (10:00–14:00 h), and evening (14:00–19:00 h). Each species was classified into one of these categories if at least 50% of the records corresponded to that time. If the percentage was less than 50%, we considered it cathemeral (active throughout the day). No separate time zone was designated for nighttime observations, as accurately identifying individual bird species captured at night was challenging. For the analysis of time zones, species with fewer than 10 samples were not considered to reflect time zone variability and accurately identify specific time zone patterns within the data. Regarding the overlap index, analysis was only conducted for species with a sample size of 50 or more to enhance the reliability of overlapping activity patterns between two species. We recorded bird activity patterns every hour, identifying and recording each bird species in the dataset. Observations for each Passeriforme were counted throughout the study period, which was further segmented into the four seasons to analyze temporal variances. To calculate the degree of temporal overlap in bird activity among different species, we divided the study period into one-hour intervals. The number of observations for each bird species within each interval was tabulated. The temporal overlap index (TOI) between each pair of bird species was then calculated as the sum of the minimum counts in each one-hour interval divided by the sum of the maximum counts [30]. This resulted in a TOI score (∆) ranging from 0 to 1, with higher scores indicating greater temporal overlap in bird activity [31]. Statistical analyses were performed using R version 4.0.2 [32]. The ‘table’ function was employed to count the number of observations for each bird species, and the ‘cut’ function was used to divide the study period into one-hour intervals. We used 1000 replicates in the bootstrap method to calculate the 95% confidence interval ∆. The temporal overlap index was calculated using the ‘pmin’ and ‘pmax’ functions in R.4.0.2 [33].

### 2.4. Analysis of Habitat Usage 

We first collected the observed usage data for each habitat type and the relative area data for these habitats. We then calculated the total observations, which is the sum of the observed usages across all habitat types. Based on this, we calculated the expected usage for each habitat type by multiplying the total observations with the relative area for each type. Next, we determined the actual proportion of usage (P_i) for each habitat by dividing the observed usage by the total number of observations [34]. To assess the variability and reliability of these proportions, we calculated the standard error (SE) for each P_i. To account for the multiple comparisons inherent in analyzing multiple habitat types, we employed the Bonferroni method [35]. We calculated the Chi-Square statistic to quantitatively assess the match between expected and actual usage. Additionally, we determined the preference for each habitat type by individual species, which was calculated as the number of observations within each habitat type divided by the total number of observations for that species. This calculation yielded a proportion ranging from 0 to 1, indicating the relative frequency of habitat use.

### 2.5. Analysis of Observation Frequency and Population Dynamics

In this study, we investigated the relationship between species observation frequency, body size, and population density across various habitats (Appendix A). Body size measurements were confined to the averaged size of a mature individual’s total length. The population density characteristics were derived from video-captured data: We calculated the low density as ≤25th percentile, medium density as >25th percentile and ≤75th percentile, and high density as >75th percentile of the ‘Individual’ column in the dataset to establish thresholds for density levels [36]. These thresholds were used to classify the average number of individuals for each species into three density categories: ‘Low’, ‘Medium’, and ‘High’. To discern the influence of body size and population characteristics on the total habitat usage by species, we applied a multiple linear regression analysis using Generalized Linear Models (GLMs) [37,38]. The dependent variable, the observation frequency of habitat usage determined by the number of camera trapping events, was defined as the cumulative number of habitats utilized by each species. The explanatory variables included body size and population characteristics, which were categorized as ‘high’, ‘medium’, or ‘low’. Given the count nature of our dependent variable, we opted for a Poisson distribution GLM with a log link function, as it is traditionally suited for modeling count data. Model fit was assessed by examining deviance residuals and Akaike Information Criterion (AIC) values, with lower AIC values indicating a better model fit to the data. The significance of model coefficients was determined using z-values and corresponding *p*-values. A coefficient was considered statistically significant if its *p*-value was below the conventional alpha level of 0.05. Statistical analyses were conducted using R version 4.0.2.

## 3. Results

### 3.1. Monitoring and Circadian Rhythms of Passeriformes

Our ecological investigation, complemented by camera trap sampling, yielded 5322 photographs, out of which 1427 were identified as independent bird sightings from 26 families and 49 species (Appendix A). Among these, a total of 13 families and 22 species of the order Passeriformes were included in the analysis of temporal activity (Table 1). Regarding temporal activity patterns, *Cyanoptila cyanomelana* and *Horornis canturians* exhibited morning activity, while *Muscicapa dauurica*, *Phoenicurus auroreus*, *Turdus hortulorum*, *Aegithalos caudatus*, and *Emberiza cioides* showed heightened activity during the afternoon. Interestingly, although a substantial number of species were observed around midday, they did not constitute more than 50% of the sightings, preventing their classification as midday species. The remaining 15 species were categorized as cathemeral, displaying activity throughout the day.

### 3.2. Habitat Usage Patterns and Population Density Influences 

As a result of habitat usage statistical analysis, the Chi-Square test yielded a value of χ^2^ = 251.57, with a *p*-value of *p* < 2.2 × 10^−16^, indicating a significant difference between the observed and expected habitat usage (Table 2). Wetlands showed a significant usage disparity with the highest proportion of use (P_i = 0.398) against its small area, a strong indicator of preference. Grasslands, conversely, displayed a lower utilization than expected (P_i = 0.181) despite a large area, suggesting underuse. Shrub areas had a higher observed use (P_i = 0.186) than grasslands, aligning closely with their proportionate availability. Trees, while encompassing the most considerable area, showed only a moderate preference (P_i = 0.235). 

The GLM analysis in this study evaluated the impact of body size and population characteristics on habitat usage (Table 3). For body size, the estimate of 0.001 is not statistically significant (*p* = 0.596), which suggests that the impact of body size on habitat usage is not statistically confirmed. Low density presented a highly significant negative estimate (−1.154), indicating that individuals with a low population use the habitat (habitat usage) significantly less compared with those with high-population characteristics. Medium density presented a statistically non-significant negative estimate (−0.099), suggesting no significant difference in habitat usage between individuals with medium and high population densities.

The study also examined habitat preferences across bird species, revealing distinct predilections for different habitat types (Figure 2). In grassland habitats, *C. sinica* emerged as a prominent species with a strong affinity for this environment. Other species, such as *S. varius* and *E. elegans*, were also observed to favor grassland habitats. *C. macrorhynchos* and *M. alba*, though less dominant, had a notable presence in these areas. In shrub habitats, *T. atrocaudata* dominated, being predominantly observed in this type of habitat. *H. amaurotis* was also commonly found in shrubs, while *Corvus macrorhynchos* had a minimal presence, and *M. alba* had no records in shrub habitats. Tree habitats appealed significantly to species such as *T. pallidus*, *P. major*, and *F. zanthopygia*, which displayed a marked preference for these environments. *G. glandarius* demonstrated a modest inclination toward tree habitats, while *M. alba*, *M. cinerea*, and *T. atrocaudata* showed no significant presence in these habitats. Regarding wetland habitats, both *M. alba* and *M. cinerea* exhibited a strong preference. Other species, including *C. macrorhynchos*, *E. elegans*, *H. amaurotis*, and *S. varius*, also favored a preference for wetland habitats.

### 3.3. Seasonal Variations in Activity Patterns and Temporal Overlap of Bird Species

Our analysis revealed significant seasonal variations in the activity patterns (Appendix A) of *C. sinica* and *E. elegans*, demonstrating overlap coefficients of 0.53 in spring, 0.79 in summer, 0.81 in autumn, and 0.23 in winter. *C. macrorhynchos* and *G. glandarius* had overlap coefficients of 0.57 in spring, 0.27 in summer, 0.35 in autumn, and 0.08 in winter (Figure 3). For *F. zanthopygia* and *T. atrocaudata*, only *F. zanthopygia* demonstrated activity in the spring; no activity patterns were detected in winter for both species, while coefficients of 0.44 and 0.33 were observed in summer and autumn, respectively. *H. amaurotis* and *T. pallidus* exhibited an overlap coefficient of 0.72 in spring, with summer and autumn coefficients of 0.52 and 0.63, respectively. Winter activity patterns were not observed for *T. pallidus*. *M. alba* and *M. cinerea* had an activity overlap in spring, with a coefficient of 0.71; no overlaps were calculated for summer, autumn, or winter due to the absence of activity. The seasonal activity of *P. major* and *S. varius* revealed overlap coefficients of 0.52 for spring, 0.44 for summer, 0.81 for autumn, and 0.69 for winter.

## 4. Discussion

Passeriformes, commonly known as perching birds, typically exhibit heightened activity during the early morning hours, a behavior referred to as the ‘dawn chorus’ [14]. This phenomenon is primarily influenced by three key factors: mate attraction and territory defense, light levels, and energy efficiency [15]. Among the species studied, *P. major* and *Horornis canturians* demonstrated pronounced morning activity patterns, aligning with the widely observed avian behavior of the dawn chorus. These species appear to take advantage of the early morning hours for foraging and other activities, likely due to the optimal conditions provided by the quiet morning environment, suitable light levels, and the need to replenish overnight-depleted energy reserves [39,40]. Conversely, five species, *M. dauurica*, *P. auroreus*, *T. hortulorum*, *A. caudatus*, and *E. cioides*, displayed heightened activity levels during the afternoon. This suggests variations in species-specific behavior, possibly driven by factors such as prey availability, predator activity, and temperature fluctuations [41]. The remaining 15 species, including *P. major* and *G. glandarius*, were classified as cathemeral, with their activities spread relatively evenly throughout the day. These species did not exhibit a distinct preference for a particular time zone, suggesting their adaptability to utilize resources and engage in activities regardless of the time [42,43].

Species with high population densities may be less constrained by competition for resources or may be more effective at exploiting those available, leading to more consistent activity patterns that increase the likelihood of observation.

In habitat usage, statistical analysis, as indicated by the Chi-Square test results (χ² = 251.57, *p* < 2.2 × 10^−16^), reveals significant differences in habitat preferences among species. Despite their smaller area, wetlands showed a disproportionately high usage (P_i = 0.398), suggesting a strong preference for this habitat. Conversely, grasslands were underutilized (P_i = 0.181) relative to their availability. This pattern implies that specific habitats, such as wetlands and shrub areas (P_i = 0.186), are more critical for these species than previously understood [44].

Additionally, GLM analysis sheds light on the influence of body size and population characteristics on habitat usage. The analysis indicates that body size does not significantly impact habitat choice (*p* = 0.596), suggesting that factors other than physical size are more pivotal in determining habitat preferences. Species with lower population densities used habitats significantly less than those with higher densities, highlighting the crucial role of population characteristics in shaping habitat utilization patterns.

The observed seasonal activity overlaps, with coefficients ranging from 0.08 to 0.81, might not only reflect behavioral flexibility but also correlate with habitat preferences as indicated by our habitat usage analysis. The high degree of overlap among species, in particular habitats during specific seasons, suggests that these habitats provide vital resources critical for survival and reproduction.

The observed seasonal overlap coefficients (Δ), ranging from 0.08 to 0.81, indicate a substantial variation in activity overlaps across different seasons, which might not only reflect behavioral flexibility but also correlate with habitat preferences, as revealed by our habitat usage analysis. For instance, the high degree of overlap (>50%) among species with similar ecological behaviors, such as *C. sinica* and *E. elegans* in spring, summer, and autumn, underscores the role of seasonality in shaping these patterns. This suggests that specific habitats provide essential resources for survival and reproduction during these times. Contrary to our initial anticipation of similar activity patterns among Passeriformes due to their shared behavioral ecology [45,46], these results show that despite their ecological similarities, these species exhibit peak activities at different times of the day. This leads to varying degrees of overlap in their activity patterns [47], demonstrating the high behavioral flexibility many passerine species display, adjusting their activity patterns according to environmental conditions [48]. 

In the context of seasonal adaptations, our observations present an intriguing picture of ecological dynamics, particularly in spring and autumn. For instance, in spring, *M. alba* and *M. cinerea* exhibit a significant overlap in activity (Δ = 0.71), which might initially suggest competition for post-winter resources like insects and early blooming vegetation. However, a more nuanced interpretation reveals this as a case of temporal niche partitioning, where these species subtly exploit the same resources but at different times or within different microhabitats. For example, one species may forage in the early hours of the morning while the other may prefer late morning, or they could select different parts of the same plant or prey on different insect species [49,50]. This form of partitioning effectively reduces direct competition, facilitating their coexistence in periods abundant with resources. Similarly, the moderate overlap between *C. sinica* and *E. elegans* (Δ = 0.53) during the same season suggests some shared ecological niche elements. However, this does not necessarily translate to direct competition. Instead, it could reflect evolved strategies for exploiting different components of shared resources or engaging in distinct activities within the same habitat. Such subtle niche differentiation might involve each species specializing in certain prey types or employing unique foraging methods, allowing them to inhabit the same area without intense rivalry [51]. When transitioning to autumn, the scenario changes for species like *F. zanthopygia* and *T. atrocaudata*, where a decreased overlap (Δ = 0.33) indicates a shift toward more pronounced niche differentiation. With resources dwindling, these species may diversify their diets and foraging tactics, targeting different food sources or using the same resources in alternative manners like varying their foraging times or exploiting different habitat sections. This adaptive flexibility is a testament to their ability to respond to the pressures of changing resource availability and competition, underlining the dynamic nature of their ecological relationships and survival strategies [52].

In the context of thermoregulation, the observed overlap indices in species pairs such as *P. major* and *S. varius* (Δ = 0.52 in spring, Δ = 0.44 in summer) and *H. amaurotis* and *T. pallidus* (Δ = 0.72 in autumn) offer a compelling glimpse into the role of thermoregulation in shaping activity patterns among these birds. In the warmer spring and summer months, the moderate activity overlap between *P. major* and *S. varius* may not just be a random pattern. Instead, it could suggest a strategic adaptation where these species align their peak foraging activities to cooler parts of the day. Such behavior appears to be a dual-purpose strategy: on the one hand, it conserves energy that might otherwise be spent on regulating body temperature under warmer conditions; on the other hand, it helps maintain the optimal body temperatures crucial for physiological processes [53]. These thermoregulatory behaviors align with the foundational principles of behavioral ecology, which emphasize the importance of balancing energy expenditure against the demands of maintaining physiological homeostasis in varied environmental conditions [54,55]. The shift in the overlap index from autumn to winter (Δ = 0.52 for *H. amaurotis* and *T. pallidus*) further illustrates this point. This change could signify a flexible behavioral adaptation to the extreme temperature characteristic of these seasons, potentially indicating strategies like basking in the sun during warmer daytime periods or seeking shelter when it gets colder [56]. Such adaptations highlight a dynamic response to the environmental challenges faced by these species. Moreover, these patterns of thermoregulatory behavior underscore the intricate ways in which animals interact with their environment. It is not just about the physical activities they engage in but also the underlying physiological needs driving these behaviors. This aspect of animal ecology is crucial, as it adds another layer of understanding to how species survive and thrive in their natural habitats. While these interpretations are consistent with current theories in avian behavioral ecology, they also open avenues for further research. It suggests that understanding the full spectrum of animal behaviors requires considering not just the visible actions but also the physiological imperatives behind them.

In the context of social structure on habitat usage, the observed differences in habitat utilization related to population density, particularly in species such as *C. macrorhynchos* and *G. glandarius*, bring to light the possible role of social dynamics in environmental interactions. These variations, marked by differing degrees of overlap across seasons, may reflect shifts in group behaviors and resource requirements [57,58]. For example, the more social or gregarious nature of *C. macrorhynchos* could result in distinct habitat usage patterns compared to species with more solitary tendencies. Such a pattern implies that social structures could influence how species exploit their habitats, potentially affecting everything from foraging strategies to shelter selection. However, while these observations are suggestive, they should be cautiously approached. The complexity of the relationship between social structures and habitat use cannot be overstated, and there are inherent challenges in drawing definitive conclusions from the available data. For one, social behaviors are multifaceted and can vary significantly even within a species, influenced by factors ranging from the immediate environmental conditions to the genetic makeup of individual animals [59]. Furthermore, while indicative, the data on habitat use and population density might not fully capture the nuances of social dynamics and their ecological impact. Given these considerations, the link between social structure and habitat use in these species remains a hypothesis that requires further exploration. It would benefit future studies to delve deeper into this aspect, possibly by employing longitudinal observations and integrating behavioral studies with ecological data. Such research could provide a more comprehensive understanding of how social interactions shape habitat utilization patterns and, conversely, how environmental pressures influence social structures within avian communities. Overall, this study highlights significant overlaps in activity patterns between specific pairs of species within Passeriformes. The individual activity trends emphasize the diversity of temporal niches occupied by these birds, offering insights into their behavioral ecology. However, it is crucial to recognize the limitations inherent in using camera traps for studying bird behavior. Camera traps, while effective for detecting presence and activity, may not capture the full range of complex behaviors exhibited by birds [60]. They are limited in their field of view and may miss aerial or high-canopy activities, which are essential aspects of avian life. Furthermore, camera traps can only record a fraction of the available habitat at any given time, potentially leading to the underrepresentation of certain behaviors or species. In addition to the constraints posed by the camera traps, other factors such as limited sample sizes and the influence of various environmental conditions such as temperature, precipitation, wind, humidity, and human presence can impact the generalizability of our findings. The circadian rhythms of birds and their behavioral responses are highly sensitive to these environmental factors, and our study only provides a snapshot of these complex dynamics [61].

Therefore, while this study contributes to our understanding of niche separation within Passeriformes in the Halla Mountain Wetland, further research is necessary to develop a more comprehensive understanding. Future studies should consider integrating camera trap data with other methodologies, such as direct observation, acoustic monitoring, and ecological modeling, to capture a broader spectrum of avian behaviors and interactions [12,62]. This multi-faceted approach would allow for a more nuanced and complete picture of the ecological dynamics within these bird communities.

## 5. Conclusions

This study, using camera trap data encompassing 5322 photographs, has contributed to a deeper understanding of the behavioral ecology of Passeriformes in the Halla Mountain Wetland. By analyzing 1427 independent bird sightings across 26 families and 49 species, we have confirmed temporal activity patterns and distinct habitat preferences, highlighting the adaptability of these species to their environment. Our observations show that species like *Cyanoptila cyanomelana* and *Horornis canturians* are predominantly active in the morning, aligning with the ‘dawn chorus’ phenomenon, while others such as *Muscicapa dauurica* and *Phoenicurus auroreus* tend to be more active in the afternoon. The wide range of temporal activities, including the cathemeral behavior noted in 15 species, underscores the behavioral flexibility prevalent within Passeriformes.

The habitat usage analysis, guided by Chi-Square and GLM statistical methods, revealed significant differences in habitat preferences among these species. The findings suggest that wetlands, though limited in area, are highly favored, in contrast to grasslands, which are relatively underutilized despite their larger area. This reflects a complex interaction between species and their habitats that goes beyond mere spatial considerations.

Moreover, this study identified the crucial role of population characteristics in influencing habitat utilization. Species with higher population densities tend to exhibit more consistent activity patterns, suggesting a correlation between population dynamics and ecological adaptability.

Our analysis of seasonal activity overlaps has revealed patterns that indicate potential resource utilization and thermoregulation strategies among these species. These observations, reflecting varying degrees of overlap across seasons, point to the ecological resilience and adaptability of the species, which are influenced by a range of factors, including social dynamics and environmental conditions.

Recognizing the limitations of using camera traps in studying avian behavior is crucial. While they detect presence and activity efficiently, camera traps may not fully capture the complete spectrum of complex avian behaviors and interactions due to their fixed field of view and intermittent recording capability. Additionally, external environmental factors such as temperature, precipitation, wind, and human presence can affect the birds’ circadian rhythms and behavior.

Given these limitations, future research should aim to integrate camera trap data with other methods, such as direct observation, acoustic monitoring, and ecological modeling. This comprehensive approach would facilitate a more complete understanding of avian behavioral ecology by capturing a wider range of behaviors and interactions.

In conclusion, this study has contributed to confirming the intricate dynamics of niche separation within Passeriformes, revealing a dynamic and adaptive avian community within the Halla Mountain Wetland. These findings provide a foundation for further research, enhancing our comprehension of the ecological roles of these species and supporting the broader efforts in wildlife conservation and management.

## Figures and Tables

**Figure 1 animals-14-00724-f001:**
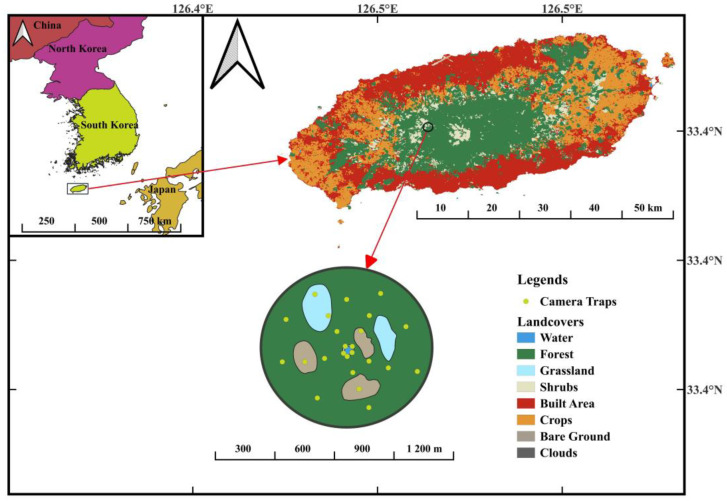
Study site location on Mt. Halla, Jeju Island, showing infrared-triggered camera installations and a schematic diagram illustrating the different types of habitats with 24 camera trapping locations.

**Figure 2 animals-14-00724-f002:**
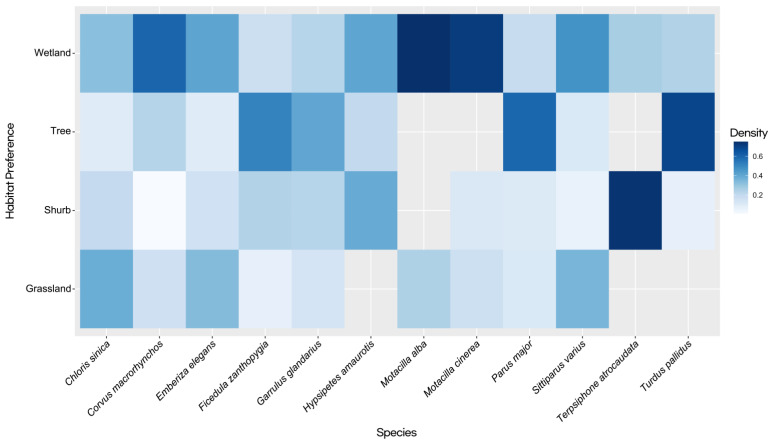
Comparison of habitat preference at camera trapping sites for each species indicated by color intensity (a deeper color represents a stronger habitat preference).

**Figure 3 animals-14-00724-f003:**
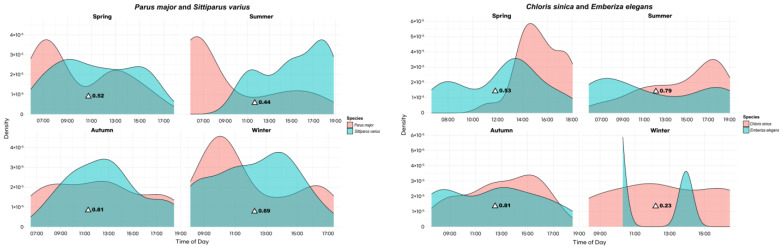
Seasonal activity patterns and temporal overlap among Passeriformes captured through camera trapping across the entire study sites (△ indicates overlap coefficients).

**Table 1 animals-14-00724-t001:** Characteristics of the 22 recorded bird species, time categorization, and migration pattern on Halla Mountain Wetland, Jeju Island, South Korea.

Family	Scientific Name	Time Categorization ^a^	Migration Pattern ^b^
Zosteropidae	*Zosterops japonicus*	C	Res
Fringillidae	*Chloris sinica*	C	Res
Tudidae	*Turdus pallidus*	C	Res
	*Turdus hortulorum*	E	SV
Phylloscopidae	*Phylloscopus xanthodryas*	C	SV
Cettidae	*Horornis diphone*	M	Res
Pycnonotidae	*Hypsipetes amaurotis*	C	Res
Paridae	*Sittiparus varius*	C	Res
	*Parus major*	C	Res
Musicapidae	*Phoenicurus auroreus*	E	Res
	*Muscicapa dauurica*	E	PM
	*Tarsiger cyanurus*	C	PM
	*Ficedula zanthopygia*	C	SV
	*Cyanoptila cyanomelana*	M	SV
Motacillidae	*Motacilla cinerea*	C	Res
	*Motacilla alba*	C	Res
Monarchidae	*Terpsiphone atrocaudata*	C	SV
Emberizidae	*Emberiza elegans*	C	Res
	*Emberiza cioides*	E	Res
Corvidae	*Garrulus glandarius*	C	Res
	*Corvus macrorhynchos*	C	Res
Aegithalidae	*Aegithalos caudatus*	E	Res

^a^: Time categorization: M = morning (6:00~10:00), A = afternoon(10:00~14:00), E = evening (14:00~18:00), C = cathemeral (the percentage was less than 50% in the time zone). ^b^: Migration pattern: Res: resident, SV: summer visitor, PM: passage migrant.

**Table 2 animals-14-00724-t002:** Comparative analysis of habitat usage in relation to area and bird observational data.

Habitat	Total Area (ha)	Po ^a^ (ha)	Observed Usage ^b^	Expected Usage ^c^	P_i ^d^	Bonferroni CI for P_i ^e^
Wetland	0.1748	0.03927	452	44.61	0.398	0.362 ≤ P_i ≤ 0.434
Grassland	5.4825	0.03927	206	44.61	0.181	0.153 ≤ P_i ≤ 0.210
Shrub	5.2049	0.04712	211	55.53	0.186	0.157 ≤ P_i ≤ 0.215
Tree	62.8258	0.06283	267	71.37	0.235	0.204 ≤ P_i ≤ 0.267
Total	73.688		1136	216.12		

^a^: Po is the proportion of the area, and this represents the percentage of each habitat monitored by camera traps, calculated based on a 5-meter effective radius for each camera. ^b^: Observed usage represents the actual number of times each habitat type was utilized. ^c^: Expected usage is determined by multiplying the total number of observations by the proportional area of each habitat, e.g., for wetlands, 452 (total observations) × 0.03927 (proportional area) = 44.61. ^d^: P_i represents the observed proportion of usage for each habitat, calculated as the observed usage divided by the total number of observations. ^e^: Bonferroni CI for P_i shows the range within which the true proportion of habitat usage is likely to fall, adjusted for multiple comparisons, e.g., in wetlands, it is 0.362 ≤ P_i ≤ 0.434.

**Table 3 animals-14-00724-t003:** Generalized linear model analysis on the influence of body size and population density on habitat usage frequency.

Variable	Estimate	Z Value ^a^	Pr(>|z|) ^b^
(Intercept)	4.768	55.356	<2 × 10^−16^ ***
Body size	0.001	0.529	0.596
Low density	−1.154	−10.491	<2 × 10^−16^ ***
Medium density	−0.099	−1.453	0.146
Residual deviance	356.92		
AIC	438.24		

***: Highly Significant (*p* < 0.001). ^a^: It is calculated as the estimated coefficient divided by its standard error, i.e., z = (Estimate − 0)/Std. Error. ^b^: it represents the probability of observing the data, or more extreme, by chance.

## Data Availability

All used data are included within the manuscript.

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
