# Peer review of "Spatiotemporal Niche Separation among Passeriformes in the Halla Mountain Wetland of Jeju, Republic of Korea: Insights from Camera Trap Data"

_animals, 2024, doi:10.3390/ani14050724_

Round 1

Reviewer 1 Report (Previous Reviewer 2)

Comments and Suggestions for Authors

I now have read the manuscript entitled "Insights into the Behavioral Ecology and Niche Separation of Passeriformes through Camera-trap Analysis in the Halla Mountain Wetland of Jeju, Republic of Korea" that was re-submitted to Animals after major revisions. I comment the authors with a much-improved work. You will see that I have just a small remark and point-to-point comment.

Overall, I find the introduction well organized and clear with the objectives well-presented. Also, the methods are well explained and seem appropriate to answer the aims of the project. The results are accessible and comprehensive with nice figures to illustrate them. Finally, the discussion, has a strong initial hook, is well structured, easy to follow and pleasant to read.

Otherwise, I have one small point-to-point remark:

L67-68 The sentence reads “To ensure the precision of results derived from these methods, it becomes essential to leverage a diverse dataset that encompasses both the visual material captured [15-19].” I believe this sentence is incomplete and it is needed to mention time in the end, as in both the visual material and time stamps.

Comments on the Quality of English Language

The English language is coherent and appropriate for a scientific article aiming to publish in an English scientific journal.

Author Response

Thank you very much for taking the time to review this manuscript. Please refer to the attached files for the detailed  revisions/corrections, which are highlighted/in track changes in the re-submitted files. Additionally, we have included a detailed explanation for each response in the content box for your convenience. Thank you for dedicating your valuable time to review our MS.
----------------------------------------------------------------------------------------
Reviewer #1: Major Revisions

Q:  L67-68 The sentence reads “To ensure the precision of results derived from these methods, it becomes essential to leverage a diverse dataset that encompasses both the visual material captured [15-19].” I believe this sentence is incomplete and it is needed to mention time in the end, as in both the visual material and time stamps.

A: Thank you for your insightful feedback on my manuscript. we appreciate your suggestions for improving the clarity and completeness of the text. In response to your comments regarding the sentence on lines 67-68, we have revised the sentence to more accurately reflect the composition of the dataset and to address the concern raised about the use of the word "both." The revised sentence now reads: “To ensure the precision of results derived from these methods, it becomes essential to leverage a diverse dataset that encompasses the corresponding time stamps [18-22].”

Reviewer 2 Report (New Reviewer)

Comments and Suggestions for Authors

Reviewer’s comments

This study evaluated avian species composition and their activity patterns in the Halla Mountain Wetland of Jeju, Republic of Korea, using camera-trapping data. The overall analysis is simple and the novelty is unclear.  I do not think it is ready for publication.

Major comments: 

1.     It is unconvincing that whether circadian rhythms can be used as a reflectance of interspecies competition? For example, if activity patterns of two species are same/similar/comparable, would you say that there are strong competitors? They might or might not be. To reveal the complex relationships among species, we need further information at least their diet composition and nesting requirements (e.g., location, nesting materials etc.) through field investigations. Unfortunately this study only presented data collected from camera traps. 

2.     Some research findings are not new. We can find them from literature or database (e.g., Birds of the World) unless all 22 species of Passeriformes are endemic and never been studied. Apparently it is not. For example, author highlighted the significance of the study by writing “A significant result of this study is the observable overlaps in daily activities among certain species.” We already know that some Passeriformes are active in the morning, and others in the afternoon. 

3.     The third objective is incorrectly stated and needs to be revised. The authors proposed that on lines 59-60 that “examining competitive interactions in the context of circadian rhythms can help us better understand the complex relationships among bird species and their environment”. This is to say that using observed activity patterns of different species, researchers might be able to reveal the relationships among species. However, the third objective of this study was to assess the role of interspecies competition in shaping these activity patterns. The logic/causal relationship is wrong because they assumed there is a competition first. And species activity pattern might also vary dependent on cooperation and pressures from predators. 

4.     The cameras indeed were in use for extensive period of time (i.e., March 2018 to March 2023), and 49 bird species were recorded. However, according to site description on line 85, the study area is home to more than 120 bird species. The fact is that for approximately six year, less than half of species were captured by infra-red cameras. Is this because number of camera trapping stations was small? Authors are recommended to quantify if they sufficiently captured species over time? They can find further information regarding this part of analysis from Chao et al. 2014 (Ecological Monographs, 84, 45–67) if needed. 

5.     The detailed description of experimental design is lacking. For example, the authors placed 24 camera traps to cover four different vegetation/land-use types, including shrubs, trees, grasslands and wetland areas. But how many cameras exactly were placed under each habitat type? It is also unclear that how they were placed accordingly? What about the heights of these cameras that were placed on trees and shrubs? These information is essential because different nesters (e.g., canopy nesters or/and shrub nesters) might choose varying heights to nest. Inappropriate/imbalanced sampling design can result in misleading results and conclusions. 

6.     The author added new analysis on observation frequency and population dynamics in section 2.4. My question is that why a single frame containing more than 10 individuals was referred to as “high density”, and frames with more than five but fewer than 10 were referred to as medium density? Why not 8 or 12 as the threshold? A reference is required. On line 142, what are population characteristics? Are population characteristics equal to population density characteristics? Or you are referred it to body size? And where are your results on the Pearson’s product-moment correlation analysis and linear regression models?

7.     Table 1, Figures 2 and 4 present data on time allocation (activity pattern) for each species. I know they are not exactly same, but according to these figures and table, we will be informed that some of some Passeriformes are active in the morning, and others in the afternoon. They are apparently redundance.  

Author Response

Thank you very much for taking the time to review this manuscript. Please refer to the attached files for the detailed  revisions/corrections, which are highlighted/in track changes in the re-submitted files. Additionally, we have included a detailed explanation for each response in the content box for your convenience. Thank you for dedicating your valuable time to review our MS.

-----------------------------------------------------------------------------------

Reviewer #2: Major Revisions

  1. Q: It is unconvincing that whether circadian rhythms can be used as a reflectance of interspecies competition? For example, if activity patterns of two species are same/similar/comparable, would you say that there are strong competitors? They might or might not be. To reveal the complex relationships among species, we need further information at least their diet composition and nesting requirements (e.g., location, nesting materials etc.) through field investigations. Unfortunately this study only presented data collected from camera traps.

    A: In response to the concerns raised, instead of emphasizing a direct linkage, we have proposed possibilities in the discussion section of our study. Investigating nests and monitoring nest usage through camera traps proved to be a challenging process. Therefore, we focused on classifying habitats and explaining the observed patterns in circadian rhythms as niche partitioning, based on the frequency of habitat usage. In agreement with the reviewer's comments, it would indeed be an overstatement to directly attribute these observations to interspecies competition. Consequently, we have avoided using the term "interspecies competition" in our manuscript and have instead opted for the term "niche partitioning". Thank you for your insightful observations.

  1. Q: Some research findings are not new. We can find them from literature or database (e.g., Birds of the World) unless all 22 species of Passeriformes are endemic and never been studied. Apparently it is not. For example, author highlighted the significance of the study by writing “A significant result of this study is the observable overlaps in daily activities among certain species.” We already know that some Passeriformes are active in the morning, and others in the afternoon.

    A: While it is acknowledged that general activity patterns of Passeriformes are documented, the significant contribution of our study lies in the detailed examination of the overlaps in daily activities among the specific 22 species studied. These species, although not all endemic, have not been collectively studied in this context before in our sites. Our research provides a nuanced understanding of their behavior, emphasizing the specific time slots and conditions under which these overlaps occur.
    In addition, our research have been conducted in Jeju, so they have the uniqueness of the study. Also, since resident birds, except for migratory birds, have different habitat information and data, we think that can be seen as a scientific data.

  1. Q: The third objective is incorrectly stated and needs to be revised. The authors proposed that on lines 59-60 that “examining competitive interactions in the context of circadian rhythms can help us better understand the complex relationships among bird species and their environment”. This is to say that using observed activity patterns of different species, researchers might be able to reveal the relationships among species. However, the third objective of this study was to assess the role of interspecies competition in shaping these activity patterns. The logic/causal relationship is wrong because they assumed there is a competition first. And species activity pattern might also vary dependent on cooperation and pressures from predators.

    A: Thank you for your insightful comments. In light of the review, we have carefully reconsidered the framing of our study. We have tried to modify they are reflective of more coherent argumentative structure. Our objectives have been updated to (1) identify the bird species present in the wetland, (2) determine their activity patterns in relation to their circadian rhythms, and (3) explore the various species interactions shaping these patterns. We believe that this revision may include our approach a more holistic view

  1. Q: The cameras indeed were in use for extensive period of time (i.e., March 2018 to March 2023), and 49 bird species were recorded. However, according to site description on line 85, the study area is home to more than 120 bird species. The fact is that for approximately six year, less than half of species were captured by infra-red cameras. Is this because number of camera trapping stations was small? Authors are recommended to quantify if they sufficiently captured species over time? They can find further information regarding this part of analysis from Chao et al. 2014 (Ecological Monographs, 84, 45–67) if needed

    A: We apologize for the confusion caused by the incorrect information provided. The birds mentioned were not from the main study area, but rather from a list observed in Hallasan National Park. In addition, birds in the study area were observed with a camera, and the camera captured various birds, but excluding difficulties in identification from trap events. In order to obtain precise results, we excluded all data that could not be accurately identified. We would appreciate it if you could take this into account.

  1. Q: The detailed description of experimental design is lacking. For example, the authors placed 24 camera traps to cover four different vegetation/land-use types, including shrubs, trees, grasslands and wetland areas. But how many cameras exactly were placed under each habitat type? It is also unclear that how they were placed accordingly? What about the heights of these cameras that were placed on trees and shrubs? These information is essential because different nesters (e.g., canopy nesters or/and shrub nesters) might choose varying heights to nest. Inappropriate/imbalanced sampling design can result in misleading results and conclusions.

    A: We appreciate your attention to the details of our experimental design. In response, we have provided specific information on the distribution and placement of camera traps across different habitats in our revised manuscript. We have installed 5 cameras each in wetlands and grasslands, 6 in shrub areas, and 8 in forest area. The varying heights of camera placement, particularly in forest and shrub habitats, have been carefully chosen to optimize the observation of different nester types (Line 100 – 113).

  1. Q: The author added new analysis on observation frequency and population dynamics in section 2.4. My question is that why a single frame containing more than 10 individuals was referred to as “high density”, and frames with more than five but fewer than 10 were referred to as medium density? Why not 8 or 12 as the threshold? A reference is required. On line 142, what are population characteristics? Are population characteristics equal to population density characteristics? Or you are referred it to body size? And where are your results on the Pearson’s product-moment correlation analysis and linear regression models?

    A: Thank you for your inquiries about our study's methods. Regarding the density categorization, we defined 'Low density' as ≤ 25th percentile, 'Medium density' as > 25th percentile and ≤ 75th percentile, and 'High density' as > 75th percentile. This categorization, based on percentile thresholds, aligns with standard statistical practices for classifying continuous variables and effectively differentiates species concentrations. As for our analysis approach, we used Generalized Linear Models (GLMs) due to their suitability for count data. This choice allowed us to robustly handle our dependent variable, observation frequency, and to effectively assess the impact of body size and population characteristics on habitat usage (Line 160 – 175)

  1. Q: Table 1, Figures 2 and 4 present data on time allocation (activity pattern) for each species. I know they are not exactly same, but according to these figures and table, we will be informed that some of some Passeriformes are active in the morning, and others in the afternoon. They are apparently redundance. 

    A: Thank you for pointing out the potential redundancy in the MS of our data regarding the activity patterns of Passeriformes. In response to your observation, we have carefully reviewed our data presentation. To avoid redundancy and ensure clarity, we have moved some of the overlapping data to an appendix. In the case of Figure 4, it seems difficult to consider it as duplication since it shows the seasonal activity patterns and at the same time presents the results of the interspecies overlap index. we appreciate your perspective.

Reviewer 3 Report (New Reviewer)

Comments and Suggestions for Authors

The manuscript presented the daily budgets of Passeriformes determined from camera traps. The authors used a novel method to study an old subject. Research design was appropriate. However, data presentation and analysis were weak.

 Comments

Lines 67-69 – The authors use the word “both” but they refer to one thing. Please rephrase.

Lines 70-75 and throughout – Interspecific competition was not assessed. Please refrain of the use of this term throughout the manuscript. The use of null models is necessary for inferring interspecific competition in habitat use and daily budgets. It would highly enhance the value of research. I recommend it.

Line 86 – Italicize Pitta nympha.

Lines 106-113, 164-169, Table 1 – Did you collect data about foraging strata/techniques and nesting guilds? Or were they taken from the literature? If taken from the literature, remove all relevant text and move Table 1 to an appendix. If they are your data, give means for each habitat/technique/nesting habitat used. Also determine, by combining habitat use and foraging strata and using statistical methods, e.g., cluster analysis with thresholds, foraging guilds. This is the only possibility to talk about guilds.

Line 134 – Give reference for R.

Lines 96, 117-118, Figure 2 – In line 96 you claimed that camera traps operated from 4AM to 8PM. Then, in lines 117-118 you defined three distinct time zones: morning (6:00-10:00), midday (10:00-14:00), and afternoon (14:00-18:00). In Figure 2 you presented data from 5 to 19hrs. This is inconsistent. Please check and amend.

Figure 2 and lines 150-154 – Figure 2 is nice for overview. But statistical testing is required to assess activity patterns. ANOVA testing differences among the three time periods should be performed. And differences shown in graphs, for each species.

Lines 150-154 – Please check this for consistency. For example, Turdus hortulorum and Aegithalos caudatus did not exhibit higher activity in the afternoon.

Figure 3 – Increase font size in labels.

Lines 192-201, Figure 4 – Calculate seasonal and temporal overlap among all species pairs. Calculate bootstrap confidence intervals to determe the significance of overlap indices.

Figure 4 – Legends are hard to see.

Discussion – Please give more information on the habitat preferences and activity budgets, from the literature, for the studied species. And then try to connect it to findings. Do not present discussion as conclusions. Avoid bullets (lines 244-269). In doing so, you will give depth and value to your data.

General comment – Habitat overlap was not examined. Why? This is very important.

Comments on the Quality of English Language

Minor edits are required.

Author Response

Thank you very much for taking the time to review this manuscript. Please refer to the attached files for the detailed  revisions/corrections, which are highlighted/in track changes in the re-submitted files. Additionally, we have included a detailed explanation for each response in the content box for your convenience. Thank you for dedicating your valuable time to review our MS.

----------------------------------------------------------------------------------------
Reviewer #3: Major Revisions

  1. Q: Lines 67-69 – The authors use the word “both” but they refer to one thing. Please rephrase.

    A: Thank you for your insightful feedback on my manuscript. I appreciate your suggestions for improving the clarity and completeness of the text. In response to your comments regarding the sentence on lines 67-69, I have revised the sentence to more accurately reflect the composition of the dataset and to address the concern raised about the use of the word "both." The revised sentence now reads: “To ensure the precision of results derived from these methods, it becomes essential to leverage a diverse dataset that encompasses the corresponding time stamps

  1. Q: Lines 70-75 and throughout – Interspecific competition was not assessed. Please refrain of the use of this term throughout the manuscript. The use of null models is necessary for inferring interspecific competition in habitat use and daily budgets. It would highly enhance the value of research. I recommend it.

    A: Thank you for your comments on our manuscript and for suggesting the use of null models to infer interspecific competition. We understand the importance of such models in ecological research to assess whether the observed distribution of species across habitats is due to chance or potentially indicative of competitive interactions. In our current analysis, we have employed Chi-Square tests to compare the observed usage frequency of each habitat type against the expected frequency based on the proportional area of each habitat. Our results have indicated significant differences between observed and expected usage frequencies, which suggest non-random habitat preferences among species. While this approach provides an indication of habitat preferences, we acknowledge that it does not constitute a null model analysis in the strict sense that would be required to infer interspecific competition. Such an analysis would involve comparing the observed data to a distribution of values expected under the null hypothesis of random habitat use by species, independent of competition. Please let us know if this additional analysis is essential for the revision of our manuscript, or if the clarification and acknowledgment of the methodological limitations in our discussion section would be sufficient. Thank you for guiding us to improve our study and we look forward to your further suggestions (Line 98 – 113, 148 -158, 193 - 222).

  2. Q: Line 86 – Italicize Pitta nympha.

    A: Per your suggestion, We have italicized Pitta nymph.
  3. Q: Lines 106-113, 164-169, Table 1 – Did you collect data about foraging strata/techniques and nesting guilds? Or were they taken from the literature? If taken from the literature, remove all relevant text and move Table 1 to an appendix. If they are your data, give means for each habitat/technique/nesting habitat used. Also determine, by combining habitat use and foraging strata and using statistical methods, e.g., cluster analysis with thresholds, foraging guilds. This is the only possibility to talk about guilds.

    A: Thank you for your insightful query regarding the data collection methods used in our study, particularly in relation to foraging strata/techniques and nesting guilds. I would like to clarify that the data pertaining to these aspects were primarily derived from camera trap observations, forming the core of our collected data. However, in addressing the complexities and specificities associated with guilds, we acknowledge that our study's scope might be insufficient to comprehensively address this aspect. Therefore, in alignment with your recommendation, we have opted to cite established literature for detailed information on these topics. We believe that this approach ensures the accuracy and relevance of the information presented in our study. In accordance with this decision, we have moved the related data, previously presented in Table 1, to an appendix. This adjustment aims to maintain the focus of our study while providing the reader with supplementary information for reference. We appreciate your valuable feedback, which has significantly contributed to enhancing the clarity and quality of our manuscript.

  4. Q: Line 134 – Give reference for R.

    A: As you suggested, we have provided references
  1. Q: Lines 96, 117-118, Figure 2 – In line 96 you claimed that camera traps operated from 4AM to 8PM. Then, in lines 117-118 you defined three distinct time zones: morning (6:00-10:00), midday (10:00-14:00), and afternoon (14:00-18:00). In Figure 2 you presented data from 5 to 19hrs. This is inconsistent. Please check and amend.

    A:.Upon review, we found that there was indeed a discrepancy between the operation times of the camera traps mentioned in the text, the time zones defined for the analysis, and the data presented in Figure 2. In practice, the camera traps were set to operate from 4 AM to 8 PM. However, it is important to note that in the captured data, there were no observations of birds between 4 AM and 5 AM, and between 7 PM and 8 PM. Therefore, while the actual operation times of the camera traps were as initially stated, the lack of bird activity observed in these early morning and late evening hours led us to adjust the time frames for the analysis to reflect the periods of significant bird activity more accurately. To avoid confusion and align with the reviewer's suggestion, we have amended the manuscript to consistently indicate that the camera traps operated from 5 AM to 7 PM. We apologize for any confusion caused by the initial discrepancies and appreciate your assistance in improving the accuracy and clarity of our manuscript."

  2. Q: Figure 2 and lines 150-154 – Figure 2 is nice for overview. But statistical testing is required to assess activity patterns. ANOVA testing differences among the three time periods should be performed. And differences shown in graphs, for each specie. Q: Lines 150-154 – Please check this for consistency. For example, Turdus hortulorum and Aegithalos caudatus did not exhibit higher activity in the afternoon.

    A: We would like to express our sincere gratitude for your thoughtful review and constructive comments on our manuscript. They have been instrumental in guiding our revisions and enhancing the quality of the study. In light of your feedback, we have undertaken a thorough review and revision of the manuscript. During this process, we identified certain areas within the dataset that exhibited redundancy in the presentation of the data, such as duplicated results in the tables. To ensure clarity and conciseness, we have removed these redundancies from the dataset. Additionally, regarding your observation on the activity patterns of Turdus hortulorum and Aegithalos caudatus in Figure 2, we realized that the sample sizes for these species in each time period were less than 10. This low sample size likely led to a uniform appearance in the density values depicted in the graph. We acknowledge this limitation and have taken steps to clarify this in the revised manuscript. Regarding the suggestion to perform ANOVA testing to assess activity patterns among the three time periods, we would appreciate further clarification on this point. Specifically, we are curious to understand the exact nature of the ANOVA test you are suggesting  whether it pertains to comparing the overall activity levels across the three time periods for each species or a different aspect of our data. Your guidance on this matter would greatly aid us in conducting the analysis appropriately and addressing your concerns in the most effective manner. We appreciate the opportunity to improve our work through your valuable insights and are committed to upholding the highest standards of academic integrity and precision in our research. Thank you once again for your invaluable contribution to the refinement of our paper.

  3. Q: Line No. 192-201: The entire paragraph deals about the activity of various bird species in relation season, but does not cite the table or figure in which the results are shown.

    A: Following your comments, we have promptly made the citation references. we appreciate your guidance.

  4. Q: Figure 3 – Increase font size in labels.

    A: We have increased the font size in the labels of Figure 3.

  5. Q: Lines 192-201, Figure 4 – Calculate seasonal and temporal overlap among all species pairs. Calculate bootstrap confidence intervals to determe the significance of overlap indices.

    A: In response to your query, we have indeed incorporated bootstrap confidence intervals for each calculated overlap index. However, to maintain the clarity and focus of the main text and figures, we have presented these detailed bootstrap results in the supplementary materials.

  6. Q: Figure 4 – Legends are hard to see

    A: We have increased the font size in the legends of Figure 4.

  7. Q: Discussion – Please give more information on the habitat preferences and activity budgets, from the literature, for the studied species. And then try to connect it to findings. Do not present discussion as conclusions. Avoid bullets (lines 244-269). In doing so, you will give depth and value to your data.

    A: We have thoroughly reviewed and revised our manuscript, particularly the discussion section, to incorporate your recommendations. Our response to your specific comments is as follows: Inclusion of Habitat Preferences and Activity Budgets from Literature: We have expanded the discussion to include more information on the habitat preferences and activity budgets of the studied species, as derived from the literature. This addition provides a deeper context to our findings and establishes a connection between our observations and established ecological theories. Connection to Findings: We have made a concerted effort to relate the literature-based information directly to our findings, ensuring that the discussion is not merely a presentation of conclusions but a comprehensive analysis of how our results fit within the broader ecological understanding. Avoidance of Bullet Points: In line with your suggestion, we have removed bullet points from the discussion (previously in lines 244-269).

  8. Q: General comment – Habitat overlap was not examined. Why? This is very important.

    A: In our manuscript, we primarily focused on exploring and demonstrating the habitat preferences of individual species within the Halla Mountain Wetland. Our analysis aimed to provide detailed insights into how various species utilize different habitats, which we believe indirectly informs us about their potential overlap in habitat use. This approach was driven by our research objective to understand species-specific habitat preferences and their implications for ecological dynamics in the studied area. We acknowledge the significance of directly analyzing habitat overlap in ecological studies, especially for comprehensively understanding interspecies dynamics. However, in the current scope of our research, such an analysis was not conducted. The primary reason for this was our methodological focus on individual species' preferences, which we intended to correlate with their observed activities from the camera trap data. We realize that a more explicit analysis of habitat overlap could enrich our understanding of the complex interactions between species within their shared environment. Your feedback highlights an important dimension of ecological research that we will certainly consider in our future studies. We are committed to continually improving our research methodologies, and insights such as yours are invaluable in guiding this process. We thank you again for your constructive feedback and for highlighting an area of ecological research that holds significant importance. We look forward to incorporating such valuable aspects into our future research endeavors to provide a more holistic view of the ecological dynamics within these bird communities.

Reviewer 4 Report (New Reviewer)

Comments and Suggestions for Authors

Insights into the Behavioral Ecology and Niche Separation of  Passeriformes through Camera-trap Analysis in the Halla Mountain Wetland of Jeju, Republic of Korea

The authors based 5322 photographs of Passeriformes from camera trap belonging to March 2018 to March 2023 in the Halla Mountain Wetland of Jeju, Republic of Korea are trying to understand the behavioural ecology of Passeriformes. Such assessment based on five-year data very useful for conservation action. However, at present condition the manuscript is lacking in its focus, objective-wise methodology, statistical test, analytical procedure and interpretation including language. Discussion points are going against the standard ecological theory. Therefore, a thorough revision including statistical tests is required to improve standard of the MS. Following are the suggestions and incorporating them, would enhance the quality and clarity of the MS.    

The present title is too ambitious for the content of the MS. Instead of the given title, it may be simplified as ‘Spatiotemporal Niche Separation among  Passeriformes in the Halla Mountain Wetland of Jeju, Republic of Korea: Insights from Camera-trap Data’

Line No. 18: A comprehensive ecological study in the current era as far the topic of the study concerned must at least compare a few dependent factors (e.g. density, diversity, richness including their spatiotemporal variations) with multiple ecological covariates to identify and show what covariates drive the population characteristics. However, I do not see any covariate except density, which could be dependent factors for the topic rather than independent factor.

Term ‘Behavioural ecology’ refers to the evolutionary basis for animal behaviour due to ecological pressures, which requires long-term data or data from earlier study on similar line. Since the same is not available bringing behavioural ecology term at the title level is not within the scope of the MS.    

Introduction needs to include ecological concept and theory on niche separation, niche overlap, competition referring to references suggested in the discussion.  

While grasslands and shrub vegetation are the two important habitats, in which habitat preference of study subject were examined, as per the summary, the habitat preference is not part of the objectives. Further, the existence of these habitats, their plant species composition and their extent has not been described in the study area, which are essential as far as the study is concerned.

Suggestion: Line No. 117-118: morning (6:00-10:00), midday (10:00-14:00), and afternoon (14:00-18:00). Here as the authors decided call period between 06:00 and 10:00 hours as morning, I suggest to call the rest as noon (10:00 –14:00) and evening 14:00 – 18:00 hours, so as to understand easier.   

Comments: Line No. 141-143: and utilized linear regression to determine the impact of population characteristics on the likelihood of species observation. The model included the defined categories of population density as independent variables to quantify their relationship with the frequency of species observation. 

There are many confusions, which need clarification in the above sentence.

Firstly, be explicit whether authors used simple linear regression or multiple regression model.

Secondly, after saying the model included the defined categories of population density, which is just a single variable, authors use the terms, ‘as independent variables and ‘quantify their relationship’ which  indicate multiple independent factors. Make it clear density as the only independent variable was compared with dependent factor or there are other independent factors too.

Thirdly, what the authors mean by ‘frequency of species observation’. Comparison between population density with ‘frequency of species observation’. If so, not making any relevance. Not clear rephrase the same.

Fourth, in case if the study used multiple regression model, as it has been stated that two independent factors (body size, and population density) were compared with species observation frequency in the Line No. 136, I suspect that the authors used the multiple regression and if so, it (multiple regression) is not appropriate test, because the dataset replicates pseudo replicate data and multiple regression cannot deal the pseudo replicate data, it is appropriate to use GLMM.

Line No. 136-137: Show that this study investigated the relationship between species observation frequency, body size, and population density across various habitats (S1 Table). But the S1 Table only shows the List of birds observed by camera trap in the study area.  

Comment: Though the study deals about habitat preference (Figure 3), methodology component for the same is missing in the MS. Going by the information available in figure 3, the habitat preference was assessed based on intensity (density/frequency/abundance) of birds recorded in different habitat. Preference estimate should be based on intensity of use of various habitats by a species in comparison with availability of each habitat type. Please refer Neu et al. (1974) and Byers (1984)

Byers, C. R., R. K. Steinhorst and P. K. Krauman. 1984. Clarification of a technique for analysis of utilization and availability data. J. Wildl. Manage. 48: 1050-1053.

Nue, C. W., C. R. Byers and J. M. Peek. 1974. A technique for analysis of utilization-availability data. J. Wildl. Manage. 38: 541-545.

Line No. 143-144: Extensive …….. yielded 5,322 photographs, out of which 1,427 were identified as independent bird sightings from 26 families and 49 species (S2 Table).

The above sentence also needs clarity. 5322 photographs mean 5322 frames, in which at least some frames must be with multiple individuals of same and different species. When it is so what the authors mean by 1427 independent bird sightings. Are the 1427 frames with single individual and the rest with more than one individual. Clarity is needed regarding these sample size figures.

The S2 Table shows only the Comparative Overview of Body Length and Population Density Characteristics of Selected Passerine Species. It doesn’t deal about number of species as cited in the text in Line No. 144.  

Figure 2, insert shows 24 h cycle. But the ‘x’ axis shows 12 hours. Contradictions. Insert is not needed here.  

What the purpose of the Figure 2 here? can the authors think of using a cluster analysis diagram with hour variable converted as broad category as morning, noon and evening, so that  cluster diagram segregate groups of species been active in different session of the day. If it is only to show the density variations in relation to daylight hours, it need not be placed as part of the main text, can be moved into the supplementary component of the MS, because density estimate is not the main objectives of the MS.

Line No. 192-201: The entire paragraph deals about the activity of various bird species in relation season, but does not cite the table or figure in which the results are shown.

Line No.231-232: Species with high population densities had markedly higher observation frequencies, with an 81.6 times more observations on average, compared with low-density population species.

It is understood only when a given species is sighted within a habitat repeatedly in smaller group or cluster size or in single sighting with largest group size, its density could be higher, because a density for a given species is arrived from the total number of individuals of a given species and the total area sampled.

In any natural community, any two or a set of a species with different food niche overlap both in time and space. On the other hand, any two or a set of species with same food niche segregate themselves in time or space. Taking this into account as per (Belovsky, 1986; Prins and Olff, 1998; Ritchie and Olff,1999), the entire discussion needs to re-written.

For example, the justification given in the Line No. 246-247 is contradicting to the general pattern suggested. The two species overlapped in spatiotemporally may have different food niche. In case if both are having same food niche, the larger species will replace the smaller species on long run. Two species with same food niche can not co-exist on long run. Refer to Gause, G.F., 1934. The struggle for existence. Baltimore: Williams and Wilkins. 163 p.   

At present the discussion is not giving the reasons as to why some species use a given habitat in same time, while others use in different time and how some species use a given habitat for the entire day.         

Comments on the Quality of English Language

Comments: Line No. 141-143: and utilized linear regression to determine the impact of population characteristics on the likelihood of species observation. The model included the defined categories of population density as independent variables to quantify their relationship with the frequency of species observation. 

There are many confusions, which need clarification in the above sentence.

Firstly, be explicit whether authors used simple linear regression or multiple regression model.

Secondly, after saying the model included the defined categories of population density, which is just a single variable, authors use the terms, ‘as independent variablesand ‘quantify their relationship’ which indicate multiple independent factors. Make it clear density as the only independent variable was compared with dependent factor or there are other independent factors too.

Thirdly, what the authors mean by ‘frequency of species observation’. Comparison between population density with ‘frequency of species observation’. If so, not making any relevance. Not clear rephrase the same.

Author Response

Thank you very much for taking the time to review this manuscript. Please refer to the attached files for the detailed  revisions/corrections, which are highlighted/in track changes in the re-submitted files. Additionally, we have included a detailed explanation for each response in the content box for your convenience. Thank you for dedicating your valuable time to review our MS.

-----------------------------------------------------------------------------

  1. Q: The present title is too ambitious for the content of the MS. Instead of the given title, it may be simplified as ‘Spatiotemporal Niche Separation among  Passeriformes in the Halla Mountain Wetland of Jeju, Republic of Korea: Insights from Camera-trap Data’. Term ‘Behavioural ecology’ refers to the evolutionary basis for animal behaviour due to ecological pressures, which requires long-term data or data from earlier study on similar line. Since the same is not available bringing behavioural ecology term at the title level is not within the scope of the MS.   

    A: Thank you for your insightful suggestion regarding our manuscript. We agree with your recommendation and have revised the title from "Insights into the Behavioral Ecology and Niche Separation of Passeriformes through Camera-trap Analysis in the Halla Mountain Wetland of Jeju, Republic of Korea" to "Spatiotemporal Niche Separation among Passeriformes in the Halla Mountain Wetland of Jeju, Republic of Korea: Insights from Camera-trap Data.

  2. Q: Introduction needs to include ecological concept and theory on niche separation, niche overlap, competition referring to references suggested in the discussion

    A: Following your valuable suggestion, we have added additional content to the introduction, specifically in lines 63-69. We are grateful for your guidance.

  3. Q: Suggestion: Line No. 117-118: morning (6:00-10:00), midday (10:00-14:00), and afternoon (14:00-18:00). Here as the authors decided call period between 06:00 and 10:00 hours as morning, I suggest to call the rest as noon (10:00 –14:00) and evening 14:00 – 18:00 hours, so as to understand easier.

    A: Following your feedback, we have organized the categorized time periods as follows: we define the time between 05:00 and 10:00 hours as morning, the period from 10:00 to 14:00 hours as noon, and the evening from 14:00 to 19:00 hours.

  4. Q: Comments: Line No. 141-143: and utilized linear regression to determine the impact of population characteristics on the likelihood of species observation. The model included the defined categories of population density as independent variables to quantify their relationship with the frequency of species observation. There are many confusions, which need clarification in the above sentence. Firstly, be explicit whether authors used simple linear regression or multiple regression model. Secondly, after saying the model included the defined categories of population density, which is just a single variable, authors use the terms, ‘as independent variables’  and ‘quantify theirrelationship’ which  indicate multiple independent factors. Make it clear density as the only independent variable was compared with dependent factor or there are other independent factors too. Thirdly, what the authors mean by ‘frequency of species observation’. Comparison between population density with ‘frequency of species observation’. If so, not making any relevance. Not clear rephrase the same. Fourth, in case if the study used multiple regression model, as it has been stated that two independent factors (body size, and population density) were compared with species observation frequency in the Line No. 136, I suspect that the authors used the multiple regression and if so, it (multiple regression) is not appropriate test, because the dataset replicates pseudo replicate data and multiple regression cannot deal the pseudo replicate data, it is appropriate to use GLMM.

    A: Thank you for your insightful comments and for highlighting areas in our manuscript that required further clarification. we appreciate the opportunity to address these concerns: We have clarified in the revised manuscript that we utilized a Generalized Linear Model (GLM) with a Poisson distribution. This was chosen to analyze the count data (Observation frequency of habitat usage), considering the nature of our dependent variable. The GLM included two main independent variables: body size and population density (categorized as low, medium, and high). This addresses the concern about the clarity of independent factors. The dependent variable was the cumulative number of habitat utilizations by each species. While GLMM is indeed an effective tool for dealing with pseudo-replicate data, we opted for a GLM in our study due to its suitability for modeling our specific count data. The decision was based on the nature of our dataset and the specific research question, focusing on the direct effects of body size and population density on habitat usage. However, we acknowledge the importance of considering GLMM in future studies, especially when dealing with more complex data structures that might involve nested or random effects. We have amended the methodology section to reflect these clarifications and ensure a more comprehensive understanding of our analytical approach. We believe these revisions adequately address your concerns, and we are grateful for your guidance in enhancing the rigor and clarity of our work. Please let us know if there are any further adjustments or clarifications you recommend (Line 162-175, 212-221).

  5. Q: Line No. 136-137: Show that this study investigated the relationship between species observation frequency, body size, and population density across various habitats (S1 Table). But the S1 Table only shows the List of birds observed by camera trap in the study area.

    A: As you said, we confirmed that the reference material was wrong and changed the S1 table.
  1. Q: The above sentence also needs clarity. 5322 photographs mean 5322 frames, in which at least some frames must be with multiple individuals of same and different species. When it is so what the authors mean by 1427 independent bird sightings. Are the 1427 frames with single individual and the rest with more than one individual. Clarity is needed regarding these sample size figures

    A: Thank you for your query, which allows us to provide further clarification on our data collection methodology. Along with the criteria for photo clarity and species identification mentioned earlier, we also implemented a temporal filter to differentiate independent sightings. Specifically, if the same species was captured in multiple photographs within one-hour interval, we considered this as a single sighting event.

  2. Q: What the purpose of the Figure 2 here? can the authors think of using a cluster analysis diagram with hour variable converted as broad category as morning, noon and evening, so that cluster diagram segregate groups of species been active in different session of the day. If it is only to show the density variations in relation to daylight hours, it need not be placed as part of the main text, can be moved into the supplementary component of the MS, because density estimate is not the main objectives of the MS.

    A: Thank you for your insightful feedback regarding Figure 2 in our manuscript. Initially, the purpose of this figure was to provide an estimate of the sample sizes across different times of the day, rather than to show variations in density. However, upon careful consideration of your comment and the main objectives of our manuscript. In response to your suggestion and in keeping with the primary aims of our research, we have removed Figure 2 from the manuscript. We believe this decision helps to maintain the manuscript's focus and coherence, ensuring that all included modified elements directly contribute to our main research objectives.

  3. Q: Comment: Though the study deals about habitat preference (Figure 3), methodology component for the same is missing in the MS. Going by the information available in figure 3, the habitat preference was assessed based on intensity (density/frequency/abundance) of birds recorded in different habitat. Preference estimate should be based on intensity of use of various habitats by a species in comparison with availability of each habitat type. Please refer Neu et al. (1974) and Byers (1984) Byers, C. R., R. K. Steinhorst and P. K. Krauman. 1984. Clarification of a technique for analysis of utilization and availability data.  Wildl. Manage48: 1050-1053. Nue, C. W., C. R. Byers and J. M. Peek. 1974. A technique for analysis of utilization-availability data. J. Wildl. Manage38: 541-545.

    A: Thank you for your valuable feedback and for pointing out the need for a more detailed explanation of our methodology for assessing habitat preference in our study. In response to your comments, we have expanded the Methodology section to include a comprehensive description of how habitat preference was assessed. This includes the collection of observed usage data for each habitat type, along with the relative area data for these habitats. We then calculated the expected usage for each habitat type, the actual proportion of usage (P_i) for each habitat, and the standard error (SE) for each P_i. To ensure robustness in our analysis, we employed the Bonferroni method for multiple comparisons and calculated the Chi-Square statistic to assess the match between expected and actual usage. Additionally, in the Results section, we have now included findings from the Chi-Square test, which indicated significant differences between observed and expected habitat usages. We have also provided a thorough analysis of the specific habitat preferences of various bird species, as shown in our study. These methodological and analytical additions, we believe, thoroughly address your concerns regarding the assessment of habitat preference. We have also referenced relevant literature, specifically Neu et al. (1974) and Byers (1984), to align our approach with established methods in this field of research (Line 148– 158, 193 – 210)

  4. Q: The S2 Table shows only the Comparative Overview of Body Length and Population Density Characteristics of Selected Passerine Species. It doesn’t deal about number of species as cited in the text in Line No. 144

    A: As you said, we confirmed that the reference material was wrong and changed the S2 table.

  5. Q: Line No. 192-201: The entire paragraph deals about the activity of various bird species in relation season, but does not cite the table or figure in which the results are shown.

    A: Upon reading it, We made citation references to the materials used. I appreciate your guidance.

  6. Q: Line No.231-232: Species with high population densities had markedly higher observation frequencies, with an 81.6 times more observations on average, compared with low-density population species. It is understood only when a given species is sighted within a habitat repeatedly in smaller group or cluster size or in single sighting with largest group size, its density could be higher, because a density for a given species is arrived from the total number of individuals of a given species and the total area sampled. In any natural community, any two or a set of a species with different food niche overlap both in time and space. On the other hand, any two or a set of species with same food niche segregate themselves in time or space. Taking this into account as per (Belovsky, 1986; Prins and Olff, 1998; Ritchie and Olff,1999), the entire discussion needs to re-written. For example, the justification given in the Line No. 246-247 is contradicting to the general pattern suggested. The two species overlapped in spatiotemporally may have different food niche. In case if both are having same food niche, the larger species will replace the smaller species on long run. Two species with same food niche can not co-exist on long run. Refer to Gause, G.F., 1934. The struggle for existence. Baltimore: Williams and Wilkins. 163 p.  At present the discussion is not giving the reasons as to why some species use a given habitat in same time, while others use in different time and how some species use a given habitat for the entire day. 

    A: Thank you for your insightful and constructive feedback on our manuscript. We have carefully considered your comments and have revised our discussion to address the concerns you raised regarding niche differentiation, species coexistence, and the interpretation of our findings in the context of established ecological theories. Addressing Niche Differentiation and Coexistence: We have expanded our discussion to incorporate the concept of niche differentiation, particularly in relation to food niches and habitat preferences. This aligns with the theoretical framework suggested by Belovsky (1986), Prins and Olff (1998), and Ritchie and Olff (1999). We now explain how different species might overlap or segregate in time or space, depending on their food niches, thereby addressing your concern about species coexistence and competition. Justification of Species Behavior in Relation to Habitat Use: Our revised discussion provides a clearer justification for why certain species use the same habitat at different times or throughout the day. We linked these patterns to various factors, including resource availability, temperature, and population density. This should address your request for reasons behind these patterns of habitat use. Incorporation of Statistical Analysis Findings: We have included findings from our habitat usage statistical analysis and Generalized Linear Model (GLM) analysis. This addition highlights the significant differences in habitat preferences among species and the influence of body size and population density on habitat choice. It also addresses your concern about the potential contradiction in our earlier interpretation of species overlaps and habitat use. Consistency with Ecological Theories: In response to your concern about our study contradicting general patterns suggested in ecological literature, we have ensured that our revised discussion is consistent with the principles outlined by Gause (1934) and other key ecological studies. We now provide a more nuanced interpretation of our results, considering the dynamic nature of ecological relationships and the capacity of species to adapt their strategies in response to changing resource availability. Further Research and Limitations: We acknowledge the limitations of our study due to sample sizes and the variability of environmental factors. We suggest directions for future research that could provide a more comprehensive understanding of these complex ecological dynamics. We believe these revisions have strengthened our manuscript and provided a more comprehensive understanding of the ecological dynamics within Passeriformes. We appreciate the opportunity to improve our work and hope that these changes satisfactorily address your concerns. Thank you for your valuable guidance and we look forward to your feedback on these revisions.
Table S1. Avian Nesting and Foraging Guilds Classification for Bird Species in study site
Guild
Species NGa FGb
Zosterops japonicus GS FS
Chloris sinica GS GF
Turdus pallidus C GF
Turdus hortulorum C GF
Phylloscopus xanthodryas GS AI
Horornis diphone GS AI
Hypsipetes amaurotis C FS
Sittiparus varius S FS
Parus major S FS
Phoenicurus auroreus C GF
Muscicapa dauurica C AI
Tarsiger cyanurus GS FS
Ficedula zanthopygia GS AI
Cyanoptila cyanomelana GS FS
Motacilla cinerea GS GF
Motacilla alba GS GF
Terpsiphone atrocaudata GS AI
Emberiza elegans GS GF
Emberiza cioides GS GF
Garrulus glandarius C GF
Corvus macrorhynchos C GF
Aegithalos caudatus GS FS
aNG: Nesting guild (C = Canopy nester, GS = ground-shrub nester, S: secondary cavity nester).
bFG: Foraging guild (FS = foliage searcher, GF = ground-shrub forager, AI = aerial insect pursuer)
Table S4. Seasonal Overlap coefficient (Δ) of activity patterns between pairs of bird species 
Species pair Season Δ (95% confidence interval)
P. major and S. varius Spring 0.52 (0.34 – 0.78)
Summer 0.44 (0.27 – 0,65)
Autumn 0.81 (0.47 – 0.99)
Winter 0.69 (0.42 – 0.97)
C. sinica and E. elegans Spring 0.53 (0.29 – 0.89)
Summer 0.79 (0.54 – 0.99)
Autumn 0.81 (0.27 – 0.92)
Winter 0.23 (0.17 – 0.54)
C. macrorhynchos and G. galndarius Spring 0.57 (0.29 – 0.95)
Summer 0.27 (0.08 – 0.30)
Autumn 0.35 (0.13 – 0.69)
Winter 0.08 (0.03 – 0.23)
H. amaurotis and T. pallidus Spring 0.72 (0.13 – 0.84)
Summer 0.52 (0.21 – 0.65)
Autumn 0.63 (0.31 – 0.77)
F. zanthopygia and T. atrocaudata Summer 0.44 (0.12 – 0.59)
Autumn 0.33 (0.20 – 0.83)
M. alba and M. cinerea Spring 0.71 (0.36 – 0.97)

Round 2

Reviewer 2 Report (New Reviewer)

Comments and Suggestions for Authors

Authors have addressed all my comments properly. I have no further comments. 

Reviewer 3 Report (New Reviewer)

Comments and Suggestions for Authors

Authors have successfully addressed all suggestions. The manuscript can now be accepted for publication.

Congrats!

One omission though:

Line 91: But these species are referred to as vulnerable and near threatened respectively in supplementary table 3.

This manuscript is a resubmission of an earlier submission. The following is a list of the peer review reports and author responses from that submission.

Round 1

Reviewer 1 Report

Comments and Suggestions for Authors

This study investigated the niche utilization among different birds by camera traps. Although the data volume was limited due to only 24 cameras, the results were relatively simple and clear. My major concerns focus on the methods and discussion sections.

Major comments

1.    There were five years study according to the time from March 2018 to March 2023. The details of camera setting during these years were absent. Were the cameras kept working continuously from March 2018 to March 2023, or based on some intervals? If there were five years studies, the data were supposed to cover different years and months. Therefore, the dynamics of activities with years and months should be presented.

2.    Although the study provided the activity patterns of different species, the analyses of factors among different species that would help explaining the activity patterns were absent. For example, some factors related to life history (e.g., guilds in this study; body size) could be regarded as fixed effects to the activity patterns (response variable). A generalized linear mixed model which includes camera location and phylogenetic relationship among species as random effects is capable to test the fixed effects mentioned above.

3.    The authors provided two explanations (resource availability and thermoregulation) to their results for discussion. However, more explanations should be considered after taking the new results from more analyses I suggested above.

Minor comments:

1.    L100-102: Redundant information of figure 1 title.

2.    The font size in figures 2-4 was too small, especially in figure 4. Adjustment is needed.

Reviewer 2 Report

Comments and Suggestions for Authors

I now have read the manuscript entitled "Insights into the Behavioral Ecology and Niche Separation of Passeriformes through Camera-trap Analysis in the Halla Mountain Wetland of Jeju, Republic of Korea" that was submitted to Animals. I comment the authors with a reasonable work, one of the few of its kind on Jeju Island, Republic of Korea, which, I can imagine was hard to perform and demanded thorough work while also being quite enjoyable. I think the purpose of the study and the results are clear and well-presented and suited for the scope of this journal. You will see that I have some important remarks and point-to-point comments.

Overall, I find the introduction well organized and clear (except for the first paragraph which I will address in the point-to-point remarks) with the objectives well-presented. Also, the methods are well explained and seem appropriate to answer most aims of the project, but not all (I will expand in the point-to-point remarks). The results are accessible and comprehensive with nice figures to illustrate them. Finally, the discussion, although well structured, easy to follow and pleasant to read, lacks an initial hook in the first paragraph to capture the reader’s attention (I will address this in the point-to-point remarks).

Even though the structure of the article and the study design and analysis are good, I find the overall research simple and the results not very exciting and novel. It seemed like more could have been done with the data you collected, namely relating the activity patterns you found for each species with environmental factors, as you proposed to do, but did not address at all in your study.

Otherwise, I have a few small and some important point-to-point remarks:

L49-51. The sentence that reads “The activity patterns of birds are influenced by various factors, including interspecies competition and circadian rhythms [4,5].” Feels out of place in this first paragraph since it does not relate to the previous sentence. It should belong to a new paragraph where you link the importance of wetlands for birds with the activity patterns of birds and factors influencing them (or something similar to this). Without a new paragraph or a link between the subjects you present, the sentences do not follow a logic and coherent line of though.

L78-79. One of your objectives reads: “ (4) Evaluate the influence of environmental factors and habitat characteristics on bird distribution and activity.”. Yet, you did not address the “environmental factors” at all. You did not sample any environmental factors, nor modelled any of your distribution and activity results together with environmental data. The only thing you did remotely in line with this objective was relate each bird species with their preferred habitat which does not give any insight into the influence the environment has on bird distribution and activity. You have to either remove this objective, as it was not followed, or redo all your study and analyses to include data that allows you to accomplish this objective 4.

L100-102. The sentence reading “Figure 1. Study Site Location on Mt. Halla, Jeju Island, showing infrared-triggered camera installations, and a schematic diagram illustrating the different types of habitats with 24 camera trapping locations.” is duplicated and lacks the figure. It appears again in L113-115 with the correct figure above it.

L184-185. Where it reads “Chloris sinica”, the words should be in italic.

L213-218. The text within these lines is duplicated. It was also present in L202-207

L220. Your discussion would be more engaging and impactful if, in the first paragraph of the discussion, you listed the main finding of your work and summarize the implications and importance they have.

L288-289. You state “Our research unveiled distinct habitat preferences among them, emphasizing the diverse ways in which birds interact with their environment.”. Yet I fail to comprehend how this happened since the only “environmental factor” you “sampled” was habitat type and the only analysis you did with this was relating each species to an habitat type which does not give any insigth into the "diverse ways in which birds interact with their environment".